# Rehabilitation Strategies for Patients with Femoral Neck Fractures in Sarcopenia: A Narrative Review

**DOI:** 10.3390/jcm9103115

**Published:** 2020-09-26

**Authors:** Marianna Avola, Giulia Rita Agata Mangano, Gianluca Testa, Sebastiano Mangano, Andrea Vescio, Vito Pavone, Michele Vecchio

**Affiliations:** 1Department of Biomedical and Biotechnological Sciences, Section of Pharmacology, University Hospital Policlinico-San Marco University of Catania, 95123 Catania, Italy; mariannaavola.md@gmail.com (M.A.); giuliarita.mangano@gmail.com (G.R.A.M.); michele.vecchio@unict.it (M.V.); 2Department of General Surgery and Medical Surgical Specialties, Section of Orthopaedics and Traumatology, University Hospital Policlinico-San Marco, University of Catania, 95123 Catania, Italy; sebymangano@hotmail.com (S.M.); andreavescio88@gmail.com (A.V.); vitopavone@hotmail.com (V.P.)

**Keywords:** sarcopenia, elderly, frailty, fractures, ageing fractures, complications, recovery, rehabilitation, nutritional supplements, physical therapy

## Abstract

Sarcopenia is defined as a syndrome characterized by progressive and generalized loss of skeletal muscle mass and strength. It has been identified as one of the most common comorbidities associated with femoral neck fracture (FNF). The aim of this review was to evaluate the impact of physical therapy on FNF patients’ function and rehabilitation. The selected articles were randomized controlled trials (RCTs), published in the last 10 years. Seven full texts were eligible for this review: three examined the impact of conventional rehabilitation and nutritional supplementation, three evaluated the effects of rehabilitation protocols compared to new methods and a study explored the intervention with erythropoietin (EPO) in sarcopenic patients with FNF and its potential effects on postoperative rehabilitation. Physical activity and dietary supplementation are the basic tools of prevention and rehabilitation of sarcopenia in elderly patients after hip surgery. The most effective physical therapy seems to be exercise of progressive resistance. Occupational therapy should be included in sarcopenic patients for its importance in cognitive rehabilitation. Erythropoietin and bisphosphonates could represent medical therapy resources.

## 1. Introduction

Sarcopenia is defined as a syndrome characterized by progressive and generalized loss of skeletal muscle mass and strength, with risk of adverse outcomes such as physical disability, poor quality of life and death. [1] Prevalence of sarcopenia varies among age groups, geographic regions and evaluated context. Estimated prevalence is 1% to 29% in community-dwelling older people, 14% to 33% in long- term care and 10% in acute hospital care populations [2,3]. Starting from 40 years of age, each ten years, healthy adults lose approximately 8% of their muscle mass. Moreover, between 40 and 70 years old, healthy adults lose an average of 24% of muscle, which accelerates to 15% per decade past the age of 70. [4] The diagnosis of sarcopenia should be based on concomitant presence of low muscle mass and low muscle function [4,5].

The European Working Group on Sarcopenia in Older People (EWGSOP) defined sarcopenia as an acute or chronic disease based on low levels three measured parameters: muscle strength, muscle quantity/quality and physical performance, as an indicator of its severity [1,6].

Early sarcopenia is characterized by size reduction in muscle. Gradually, it also occurs as a decrease in the quality of muscle tissue, which leads to loss of functionality and fragility [7,8]. The assessment of sarcopenia requires objective measurements of muscle strength and muscle mass. Several methods of evaluation for sarcopenia considered walking speed, the circumference of the calf, the analysis of bioimpedance, grip strength and DEXA imaging methods. However, none of these measurements are sufficiently sensitive or specific [8,9,10]. Sarcopenia has been identified as one of the most common comorbidities associated with femoral neck fracture (FNF) patients [11]. Among these different comorbidities, apart from Sarcopenia, protein-energy malnutrition has been reported in 20 to 85 % of cases, depending on age and gender [12,13]. Besides the age-related loss of muscle mass, trauma mechanism, and consequent associated immobilization, cause negative adjustment in body composition. Bed confinement, and the reduced mobility of hospitalized older patients, are associated with loss of muscle mass, muscle function and bone mineral density from the 10th day, and up to two months, after the fracture [13,14]. During the first year from fracture, about 5–6% of muscle mass may be lost [15,16]. It was reported that 28% of patients who were outpatients before hip fracture were unable to walk 12 months after surgery, while as many as 25–30% of patients were unable to return to their previous situation. [16,17,18].

Treatment of sarcopenia is mainly nonpharmacological. First, an adequate nutrition to ensure the intake of micronutrients and macronutrients is needed. Calories should be 24 to 36 kcal/kg per day; a minimum daily protein intake of 1.0 g/kg body weight, up to 1.5 g spread equally over three meals and maintenance of serum vitamin D levels to 100 nmol/L (40 ng/mL) from vitamin D–rich diet or vitamin D supplementation. Supplementation with creatine monohydrate, antioxidants, amino acid metabolites, omega-3 fatty acids and other compounds are being studied [3]. A crucial role is played by physical activity, especially resistance training, which is the key element for increasing muscle strength and physical performance. Currently, the strategy of combined nutrition supplementation and exercise appears encouraging in the management of sarcopenia. 

The aim of this review is to evaluate the impact of rehabilitation with or without other interventions, including nutritional supplementation and pharmacological therapy, on indicators of sarcopenia for FNF (femoral neck fracture) patients. 

## 2. Experimental Section

### 2.1. Literature Search Strategy

To find clinical trials involving the rehabilitation of sarcopenia and FNF, two authors (M.A, G.R.A.M.) searched in three medical databases (PubMed, Cochrane Library and PEDro) during the month of June 2020. The terms used for the research were sarcopenia and hip fracture and rehabilitation.

### 2.2. Selection Criteria

The selected articles had to be published in the last 10 years, written in the English language and had to be randomized controlled trials (RCT), observational studies or cases reports published in peer reviewed journals. The authors excluded articles written in other languages, studies with no results or subjects involved and reviews about the topic. Papers with no accessible data, or no available full texts, were also excluded. M.A. and G.R.A.M. selected the studies independently, resolving any discrepancies about the selection by discussion. The senior investigator (M.V.) was consulted to revise the selection process.

## 3. Results

### 3.1. Search Progress and Data Extraction

A total of 74 articles were selected based on their titles. Excluding doubles, 63 articles were screened upon their titles. At the end of this screening, 14 abstracts were selected and read independently by the two authors. Five abstracts were excluded: three were not trials, two were reviews and one did not assess sarcopenia in the subjects studied. The authors screened and read eight full texts, one of which was excluded, being a rehabilitation protocol without results on patients.

Every full text was examined, and characteristics of the study, study sample, type of rehabilitation and treatment, outcome measures and results were extracted from the full text and summarized in Table 1.

Upon the seven full texts eligible for this review, three examined the impact of conventional rehabilitation and nutritional supplementation, based on food richness of proteins and aminoacids, on patients affected by sarcopenia following FNF [16,19,20] Three papers evaluated the effects of rehabilitation protocols only, especially comparing new methods to conventional rehabilitation and evaluating the impact of sarcopenia on rehabilitation progress [21,22,23]. A study explored the intervention with erythropoietin (EPO) in sarcopenic patients with femoral intertrochanteric fractures and its potential effects on postoperative rehabilitation. [24]

### 3.2. Sarcopenia Diagnosis

Various definitions of sarcopenia have been developed by different international consensus panels, (the Asian Working Group on Sarcopenia (AWGS), the European Working Group on Sarcopenia in Older People (EWGSOP) and the Foundation of the National Institute of Health, International Working Group on Sarcopenia) each defining cut-off values from mobility limitation measures (appendicular skeletal mass index, grip strength and physical performance). In our review, the authors specifically used the diagnostic criteria included in the AWGS [21,22,23,24,25] and EWGSOP [1,16,19,20].

The EWGSOP defines sarcopenia when ASM is less than 20 kg for men and 15 kg for women, ASM/height^2^ is less than 7.0 kg/m^2^ for men and 5.0 kg/m^2^ for women (muscle quantity), grip strength is less than 27 kg for men and 16 kg for women, chair stand > 15 s for five rises (muscle strength), gait speed is no more than 0.8 m/s, short Ppysical performance battery (SPPB) is less than 8 points, timed up and go (TUG) test is less than 20 s and 400 m walk test is completed in more than 6 min or not completed at all (physical performance) [1].

AWGS criteria include decreased handgrip strength (males < 28 kg, females < 18 kg), physical performance evaluated with gait speed ≤ 0.8 m/s or 5-time chair stand test: ≥12 s or short physical performance battery: ≤9, and loss of muscle mass, indexed by appendicular skeletal muscle mass (ASM) divided by height squared evaluated through dual-energy X-ray absorptiometry (M: <7.0 kg/m2, F: <5.4 kg/m^2^) or bioelectrical impedance analysis (M: <7.0 kg/m^2^, F: <5.7 kg/m^2^) [25].

### 3.3. New Rehabilitation Protocols

In Study 1 (Table 1), the functional outcomes of a new integrated rehabilitation management (FIRM) were assessed in sarcopenic and nonsarcopenic inpatients [21]. Sixty-eight patients (32 Sarcopenic and 36 nonsarcopenic) who had undergone surgery for fragility FNF were included.

FIRM included intensive physical and occupational therapy, fall prevention education with discharge planning and referral to community-based care. After surgery, the patients stayed in hospital with 10 days of physical therapy with two sessions per day and four days of occupational therapy. Physical therapy consisted of weight bearing exercises, strengthening exercises, gait training and aerobic exercise, and functional training progressed gradually based on the individual’s functional level. Occupational therapy aimed to train the patients in ADL (transfer, sit to stand, bed mobility, dressing, self-care retraining and using adaptive equipment).

The outcome measures used in the eligible studies were walking ability through two scales: the KOVAL walking ability scale [26] and the functional ambulatory category (FAC) [27]; general mobility; balance and fall risk; cognitive functioning; quality of life; mood; ADL; frailty and handgrip strength of the patients; modified Rivermead mobility index [28]; Berg balance scale [29]; MMSE [30]; Korean version of the geriatric depression scale [31]; the Euro quality-of-life questionnaire 5-dimensional classification [32]; the Korean modified Barthel index [33]; the Korean instrumental ADL [34] and the Korean version of the fatigue, resistance, ambulation, illnesses and loss of weight (FRAIL) scale [35]) at admission to the in-hospital rehabilitation unit and at discharge.

KOVAL and FAC were significantly improved in both sarcopenic and nonsarcopenic patients. Prefacture ambulatory functioning, rather than the presence of sarcopenia, was significantly correlated with short-term recovery of ambulatory functioning. Mobility, balance, cognitive functioning and quality of life improved in both groups, demonstrating the clinical effectiveness of FIRM in sarcopenic patients. In contrast, K-IADL (*p* = 0.029) and K-FRAIL (*p* = 0.023) scores were significantly improved in only the nonsarcopenia group after rehabilitation.

Limitations of this study were the short time after which the outcomes were evaluated (after two weeks of interventions) and the exclusion of several patients before the start of the treatment. The use of the sarcopenia classification itself may have affected the group allocation. Even though the results of Study 1 suggest that FIRM was effective for short-term functional recovery in older patients with or without sarcopenia who have suffered fragility hip fracture, further research comparing FIRM with conventional therapy is needed.

Study 2 (Table 1) [22] evaluated the FIRM program in a prospective observational investigation of 80 patients (35 Sarcopenic and 45 nonsarcopenic) older than 65 after FNF surgery. The author, unlike the previous study, ruled out gait speed from the diagnostic criteria for sarcopenia in the sample evaluated because this result could not be estimated before the fracture or surgery. The FIRM program was administered during two weeks of hospital stay after surgery. All functional outcomes (KOVAL, FAC, EQ-5D, K-IADL, and K-FRAIL) were assessed on admission for rehabilitation, at discharge, and at the three and six months follow-up visits after surgery (or with a telephone interview). In the considered sample, patients with sarcopenia had impairment in cognitive function in a significantly superior percentage than the nonsarcopenic group. Both groups had improvement in the primary outcome (KOVAL) and functional outcome (FAC score) after discharge. Other evaluations, excluded HGS, significantly improved in both groups with no significant difference. Even though sarcopenia was not a predictor of poorer results in ambulatory independence, at six months from surgery, the type of operation and high HGS (handgrip strength) were significantly correlated.

Study 2 [22] demonstrated that ambulation and functional outcomes were improved in patients with or without sarcopenia suffering from fragility after FNF surgery, due to a complete multidisciplinary rehabilitation. Limitations were caused by the assessment of sarcopenia in patients soon after the surgery, namely the time of follow-up that in fragile patients may have been longer, and the lack of a control group following conventional rehabilitation.

Study 3 [23] compared the efficacy of an antigravity treadmill (AGT) combined with conventional physical therapy, and physical therapy alone, in a double-blinded (to outcome) study. Selected patients were 65–90 years old, who had undergone surgery for FNF associated with sarcopenia, according to the AWGS recommendation [25]. Thirty-eight patients included in the primary analysis were treated. One group (*n* = 19) had only standardized rehabilitation treatment for 30 min per day for 10 days, the other (*n* = 19) received standardized treatment plus AGT for 20 min per day. Standardized therapy consisted in passive hip and knee mobilization, strengthening of the hip abductor and extensor muscles, transfer, and gait training on the floor and stairs.

The outcomes evaluated were the same as Studies 1 and 2 [21,22], except for the absence of the I-ADL measurement. The experimental group experienced higher and longer therapeutic effects, with improvement in all outcomes. However, in both groups, KOVAL and FAC scores were slightly improved and then decreased from 3 three to 6 months. This study provided evidence not only that AGT with CR is more effective than only CR for sarcopenic patients, but also that there is a strong association between muscle mass and bone mass, supporting the theory that muscle forces mediate mechanical loading effects on bones [36]. Limitations of Study 3 were the high amount of drop outs after hospital discharge, it was carried out in only one center, and the number of the sample was not sufficient to significantly represent subgroups with different cognitive function, hip fracture and hip operation type.

### 3.4. Nutritional Supplements and Physical Therapy

Study 4 (Table 1) [16] was a randomized controlled study evaluating the effects of combined therapy with bisphosphonate, protein-rich nutritional supplementation and conventional rehabilitation in 79 sarcopenic patients after FNF [16]. Measured parameters were body composition, hand grip strength (HGS) and health-related quality of life (HRQoL). Patients were randomized into three treatment groups. All patients received calcium 1 g and vitamin D 800 I.E. divided into two daily doses for 12 months. The nutritional supplementation group (protein + energy = N group, *n* = 26) received a 200 mL package twice daily, each containing 20 g of protein and 300 kcal. This supplement was given for the first six months after FNF, combined with 35 mg risedronate once weekly for 12 months. The second group (B, *n* = 28) received risedronate alone, 35 mg once weekly for 12 months. The controls (C, *n* = 25) received only calcium and vitamin D for 12 months. Treatment began as soon as the patients were medically stable after surgery, able to take orally administered medications and able to sit upright for one hour after intaking bisphosphonates.

Energy supplementation combined with bisphosphonate, vitamin D and calcium had no positive effect on hand-grip strength, HRQoL, or lean mass, when compared to administration of bisphosphonate along with vitamin D and calcium supplementation, or just vitamin D and calcium supplementation, after FNF. Protein and energy supplementation combined with conventional rehabilitation was not able to preserve lean mass after a hip fracture better than vitamin D and calcium alone or combined with bisphosphonates. There were no intergroup differences concerning effects on HGS or HRQoL, but intragroup improvement in HGS, and a positive effect on HRQoL, were seen in the nutritional supplementation group. A limitation of this study was the small sample size.

In Study 5 (Table 1) [19], 32 patients (23 Sarcopenic, nine nonsarcopenic) aged more than 65 years were enrolled three months after osteoporotic FNF and treated with total hip replacement. The authors evaluated the impact of a two months rehabilitative protocol, combined with dietetic counseling with or without essential aminoacid supplementation, on functioning. Patients were divided into two groups. Patients in group A (*n* = 16, 11 Sarcopenic, five nonsarcopenic) were treated for two months with essential aminoacid supplementation sachets of 4 g per day. Furthemore, patients performed a concomitant specific physical exercise rehabilitative program consisting of five sessions of 40 min each per week for two weeks with the supervision of an experienced physiotherapist, and received dietetic counseling. The physical activity included walking training, resistance and stretching exercises and balance exercises. After these two two weeks, all participants performed a home-based exercise protocol up to the end of the study period, two months from intervention. Patients in group B (*n* = 16, 12 Sarcopenic and four nonsarcopenic) performed the same physical exercise rehabilitative program as group A and received concomitant dietetic counseling alone, without essential amino acid supplementation.

Outcome measures were the hand grip strength test (HGS), physical performance, using the timed up and go test (TUG) [37], level of assistance measured by the Iowa level of assistance scale (ILOA) [38], nutritional assessment, with evaluation of daily caloric intake and daily protein intake, and the health-related quality of life (HRQoL) evaluation. All outcome measures were assessed at baseline (T0) and after two months of treatment (T1). Patients in both groups were divided into sarcopenic and nonsarcopenic patients. All patients in both groups showed statistically significant differences in all primary outcome measures (HGS, TUG, ILOA) at the T1 evaluation (*p* < 0.017). Sarcopenic patients in group A showed statistically significant differences in all primary outcomes (HGS, TUG, ILOA) at T1 (*p* < 0.017), whereas sarcopenic patients in group B showed a significant reduction of ILOA at the end of treatment. On the other hand, in nonsarcopenic patients, they found no differences at T1 in the TUG test and level of assistance test. In both groups, there were no differences at T1 in all other outcome measurements. Furthermore, there were no differences between groups in all outcome measuresments both at baseline and after two months of treatment.

Even though it was performed on a small sample size, data emerging from this study showed a good impact of this combined intervention on function and disability in hip fracture patients after two months of treatment. Essential amino acid supplementation induced considerable improvements in the sarcopenic subpopulation of the study.

Study 6 (Table 1) [20] was a multicentric randomized trial evaluating a nutritional supplement, enriched with β-hydroxy-β-methylbutyrate (HMB), calcium (Ca) and 25-hydroxy-vitamin D (25(OH)D) during rehabilitation therapy to improve muscle mass and, thereby, functional recovery. It included 107 sarcopenic patients aged more than 65 years old with FNF. There were 15 drop-outs during the study. This was the first study to evaluate the effects of HMB in sarcopenic patients with hip fractures. Patients in the intervention group (IG, *n* = 49) received a standard diet plus oral nutritional supplementation enriched with 0.7 g/100 mL of HMB, 227 IU/100 mL of 25(OH)D and 227 mg/100 mL of Ca, while those in the control group (CG, *n* = 43) received a standard diet only. Physical therapy was based on moving patients early, using technical aids, and rehabilitation of activities of daily living including exercises to strengthen the lower limbs, balance exercises and walking retraining in individual or group 50 minute sessions, once a day five days a week. Outcomes measured were gait speed, hand grip strength, appendicular lean mass (aLM, kg/height^2^), nutritional assessment carried out by the Mini Nutritional Assessment-Short Form (MNA-SF) [39] and patients’ functional situation using the Barthel index (BI) [40] and the functional ambulation categories (FAC) score [27].

The outcome variable was the difference between aLM upon discharge and aLM upon admission (Δ-aLM). BMI and aLM were stable in intervention group (IG) patients, whilst these parameters decreased in the control group (CG). The concentration of proteins (*p* = 0.007) and vitamin D (*p* = 0.001) increased more in the IG than the CG. A positive effect of oral nutritional supplementation was reported on recovery of ADL. The recovery of ADL was more common in the intervention group (68%) than in the control group (59%) (*p* = 0.261).

This study had a number of limitations. Patients received rehabilitation five days a week. It would be interesting to see whether participation in a program of resistance exercises during the patients’ stay at a rehabilitation center improved the functional results reported. The authors could not do any follow-up of patients after discharge to assess whether the benefits obtained were maintained. Furthermore, diagnostic criteria for sarcopenia proposed by the EWGSOP were difficult to apply in patients with hip fractures admitted to rehabilitation units, because most of the patients were unable to walk when they arrived. Despite these limitations, this research had some important strengths. Due to the characteristics of the patients included, this study could be representative of the geriatric population admitted to rehabilitation centers.

### 3.5. Other Treatments

Study 7 (Table 1) [24] assessed the effects of recombinant human erythropoietin (EPO), already used in in sarcopenic patients for perioperative recovery, in patients with femoral intertrochanteric fracture and sarcopenia, to investigate its potential benefits on postoperative rehabilitation. EPO, through the activation of the signaling cascades in hematopoietic cells, may stimulate proliferation and differentiation of skeletal muscle myoblasts, making the skeletal muscle a potential target [41,42].

The effects of EPO were analyzed in 141 sarcopenic patients older than 60 years with intertrochanteric femoral fracture, randomly divided in intervention and control groups and examined by sex. The intervention group (*n* = 83) received recombinant human erythropoietin via intravenous injection once per day for 10 days after surgery. All patients, including the control group (*n* = 58) received adequate nutrition and exercise for recovery. The outcomes evaluated were: handgrip strength, appendicular skeletal muscle (ASM) index and postoperative hospitalization and infection. The intervention group, especially in female patients, had significant improvement in handgrip strength during the first week after the surgery. The improvement was consistent in the following three weeks. Even the ASM index was improved, with a more important improvement, but not significant, in the intervention group. The rate of post-operative infection and length of hospitalization were significantly decreased in patients who received EPO intervention.

## 4. Discussion

In this review, we considered seven studies of older adults (>60 years) in which both rehabilitation and nutrition, alone or combined, were used to improve recovery after hip fracture surgery in terms of walking independence, muscle strength, mobility, live activity and fragility. The studies included participants with different degrees of general, cognitive and mobility functions, who had experienced different types of fracture and undergone various surgery methods. The rehabilitation and supplementation strategies, as well as study designs (duration and setting) were different.

The main finding was that sarcopenia, being a multifactor disease, needs a treatment that cannot rely on a single drug. The treatment should be a combination of methods including nutritional intervention, intervention of functional exercise and medications [24]. Physical inactivity was negatively linked to losses of muscle mass and strength, suggesting that increasing levels of physical activity should have protective effects. Also, muscle strength is a critical component of walking, and its decrease in the elderly contributed to a high prevalence of falls [6,43]. Furthermore, early ambulation after hip fracture had beneficial effects on functioning, readmission rate and multidisciplinary rehabilitation reducing the risk of poor outcomes, such as death and admission to nursing homes following FNF [44].

To strengthen muscle and physical function, progressive resistance exercise training is a commonly used tool [21,22,23]. Ambulatory independence is a crucial outcome to examine in patients after hip-surgery, and it must be evaluated before and after the surgery intervention and rehabilitation protocol. In Study 1 [21], it was found that ambulatory independence is more associated with individual ambulatory function before the fracture than in the presence of sarcopenia. However, Study 2 [22] considered poor ambulatory independence as predictive factor for worse results in the evaluated outcome.

Progressive resistance training, associated with occupational therapy, in the above-mentioned studies, resulted in important improvements in walking ability, strength and general mobility, especially in the short-term rehabilitation of sarcopenic patients. Occupational therapy may also have an important role in cognitive function. Cognitive function is a crucial factor, affecting the rehabilitation outcomes after FNF in patients. When occupational therapy was not involved, there was no significant difference in outcome measurements between the two groups at all follow-ups in K-MMSE [21,22].

Type and intensity of exercise is an important variable that significantly influences functional outcomes in FNF patients. Study 3 [23], compared the effects of antigravity treadmill rehabilitation with conventional rehabilitation and conventional rehabilitation alone, and which did not include progressive resistance training and was uncertain in terms of compliance, found an important and significant improvement in the ability to walk, ambulatory function, general mobility, balance and quality of life in the experimental group. The antigravity treadmill, in fact, allowed a task-specific repetitive approach, facilitating the practice of numerous complex gait cycles, which were not possible in the control group.

In the literature, less is reported about the role of diet in older age, although there is evidence that improvements in diet among older adults at risk of developing sarcopenia may have the potential both to prevent, or delay, age-related losses of muscle mass and function, as well as being potential management strategies for sarcopenic patients. However, existing evidence from nutrient supplementation studies is mixed [2,6].

In our review, the effects of provision of additional amino acids, protein, bisphosphonates, calcium, Vitamin D and HMB, in combination with a standardized diet and exercise training, were reported. The supplements differed in type, dose, frequency and delivery among the patients, as did the results and improvement in patients. The sample was somewhat evenlydistributed in terms of age, sex and type of fracture. All three studies (Studies 4, 5 and 6) showed that supplemental nutrition improved functional results in patients with sarcopenic FNF. However, some findings must be discussed. Study 4 [16] did not confirm any hypothesis because the improvements were not significant between the different groups. However, in the nutritional supplementation group, analysis did show a positive effect on quality of life and handgrip strength. In the other two studies, significant improvement was seen in ADLs, in particular, and in HGS and walking ability in the intervention groups [19,20]. Moreover, Study 5 [19] found that sarcopenic patients with amino acid intake had important improvements in ADLs, compared to other groups. The same difference did not occur in the nonsarcopenic patients. The improvement disappeared after two months when the intake was suspended. This may prove the importance of amino-acid supplementation, especially in sarcopenic patients after hip surgery, beinmg maintained for a longer period in older adults.

As for medical therapy, no drugs are specifically designed for the treatment of sarcopenia. Testosterone, growth hormone and beta-adrenergic receptor agonists are commonly used to improve sarcopenia [45], but more research is needed because they do not always improve muscle function [46].

Study 7 [24] tried to include EPO as a drug to treat sarcopenia when used as a perioperative red blood cell mobilization drug in patients with FNF. The authors found that EPO improved the muscle strength of female patients with sarcopenia during the perioperative period, increased muscle mass of both women and men to a certain degree and significantly reduced the incidence of complications during the preoperative period. EPO may work as a new treatment option for patients with FNF in short-term postoperative rehabilitation.

## 5. Conclusions

Physical activity, in its various forms, and dietary supplementation, are the basic tools of prevention and rehabilitation of sarcopenia in elderly patients after hip surgery. Exercise training increases muscle mass in the elderly population with varying fragility and nutritional status, helping outpatient recovery, which is the primary outcome in these patients. The most effective physical therapy seems to be exercise of progressive resistance. However, occupational therapy should be included in sarcopenic patients for its importance in cognitive rehabilitation, especially in older adults, to help their return to normal daily activities. Nutritional support, combined with task-specific repetitive exercises, is supported by accumulating evidence for improving sarcopenia and preventing disability. Protein-rich dietary supplementation should primarily include amino acids for a long period in elderly patients. Treatment should include medical therapy, such as erythropoietin and bisphosphonates, which are increasingly important resources, even though they need further research for their validation.

## Figures and Tables

**Table 1 jcm-09-03115-t001:** Characteristics of examined studies.

Author	Study Type	Treatment	Type of Fracture	Sample Size	Outcome Measures	Results/Conclusions
Flodin et al. 2015	Randomized controlled study	40 g of protein and 600 kcal combined with risedronate and calcium 1 g and vit D 800 IE (group N); risedronate and calcium 1 g and vit D 800 IE (group B); calcium 1 g and vit D 800 IE (group C). All groups received conventional rehabilitation	Femoral neck or Trochanteric fracture	79 patients: 56 women (71 %), 23 men (29 %); Mean age 79 (SD 9, range 61–96 years)	Body composition, Hand grip strength (HGS), Health-related quality of life (HRQoL)	No differences among the groups regarding change in fat-free mass index (FFMI), HGS, or HRQoL. Intra-group analyses showed improvement of HGS between baseline and six months in the N group (*p* = 0.04). HRQoL decreased during the first year in the C and B groups (*p* = 0.03 and *p* = 0.01, respectively) but not in the nutritional supplementation N group (*p* = 0.22).
Invernizzi et al. 2018	Randomized controlled study	Two groups (A and B). Both groups performed a physical exercise rehabilitative program and received dietetic counseling; only group A was supplemented with two sachets of 4 g/day of essential amino acids.	Osteoporotic hip fracture	32 patients aged more than 65 years (mean aged 79.03 ± 7.80 years) divided in two groups. Patients in both groups were divided into sarcopenic and nonsarcopenic.	Hand grip strength test (HGS), Timed Up and Go test (TUG), Iowa Level of Assistance scale (ILOA), Nutritional assessment, Health-related quality of life (HRQoL). All the outcome measures were assessed at baseline (T0) and after two months of treatment (T1)	Sarcopenic patients in group A showed statistically significant differences in all the outcomes at T1 (*p* < 0.017), whereas sarcopenic patients in group B showed a significant reduction of ILOA only. In nonsarcopenic patients, no differences at T1 in all outcome measures.
Lim et al. 2018	Prospective Observational Study	FIRM (Fragility Fracture Integrated Rehabilitation Management)	Femoral neck, Intertrochanteric, or Subtrochanteric fracture.	68 patients; M = 15, F = 53 Sarcopenia (*n* = 32): Age 81.66 ± 7.49; Nonsarcopenia (*n* = 36): Age 79.81 ± 5.95	KOVAL, FAC, (primary) MMRI, BBS, MMSE, K-GDS, EQ-5D, K-MBI, K-IADL, K-FRAIL scale (secondary)	The primary outcomes improved significantly in both sarcopenic and nonsarcopenic patients. Mobility, balance, cognitive functioning and quality of life improved in both groups, K-IADL (*p* = 0.029) and K-FRAIL (*p* = 0.023) scores were significantly improved in only the nonsarcopenia group after rehabilitation.
Lim et al. 2019	Prospective Observational Study	FIRM (Fragility Fracture Integrated Rehabilitation Management)	Femoral neck, Intertrochanteric, or Subtrochanteric fracture.	80 patients; M = 18, F = 62 Sarcopenia (*n* = 35): Age 82.8 ± 7.5 Nonsarcopenia (*n* = 45): Age 79.7 ± 6.5	KOVAL, FAC, (primary) MMRI, BBS, MMSE, K-GDS, EQ-5D, K-MBI, K-IADL, K-FRAIL scale, HGS (secondary)	Koval and FAC scores improved over time (*p* < 0.001). The two groups did not differ in terms of the time course of improvement in Koval scores. There was no difference between the groups regarding the time course for improvement in FAC scores after discharge. All secondary functional outcomes, except for HGS, significantly improved over time in both the sarcopenia and nonsarcopenia groups, even though the functional status of the sarcopenia group was lower at both the three- and six month follow-up evaluations. However, the two groups did not differ significantly in terms of final functional status
Malafarina et al. 2017	Multicentre randomized trial	Standard diet (1500 kcal, 23.3% protein (87.4 g/day), 35.5% fat (59.3 g/day) and 41.2% carbo- hydrates (154.8 g/day) plus oral nutritional supplementation (ONS) enriched with CaHMB 0.7 g/100 mL, 25(OH)D 227 IU/100 mL and 227 mg/100 mL of calcium. (Intervention Group (IG); Standard diet only (Control group CG). All patients received Physical therapy	Various type of hip fracture	107 patients aged 65 years and over (mean age 85.4 ± 6.3, 74% female)	Body composition, Hand grip strength (HGS), Mini Nutritional Assessment—Short Form (MNA-SF), Barthel index (BI), Functional Ambulation Categories (FAC) score	BMI and lean mass were stable in IG patients, while decreased in the CG. The concentration of proteins and vitamin D increased more in the IG than in the CG. The recovery of ADL was more common in the IG (68%) than in the CG (59%) (*p* = 0.261)
Min-Kyun et al. 2020	Randomized Control Double-Blinded Trial	Antigravity Treadmill (Cxperimental Group) Conventional Rehabilitation (Control Group)	Femoral neck, Intertrochanteric, or Subtrochanteric fracture.	38 Patients; M = 12, F = 26 65–90 years old;	KOVAL (primary), FAC, BBS, Korean version of MMSE, EQ-5D, K-MBI, HGS (secondary)	Higher and longer improvement in KOVAL, FAC score, BBS, EQ-5D, and K-MBI in experimental group. The comparison of change scores in BBS between the two groups revealed a between-group difference of 11.63 (95% CI: 5.85, 17.40; *p* for trend = 0.001), 9.00 (95% CI: 2.28, 15.71; *p* for trend = 0.006), and 11.05 (95% CI: 3.62, 18.48; *p* for trend = 0.006), respectively. In the EQ-5D and KMBI, the experimental group showed an improvement of 0.49 and 32.63 scores, respectively, compared with 0.23 and 16.00, respectively, by the control group in the three weeks. The comparison of change scores in EQ-5D and K-MBI between the two groups revealed a between-group difference of 0.25 (95% CI: 0.10, 0.41; *p* for trend = 0.005) and 16.63 (95% CI: 4.80, 28.45; *p* for trend = 0.009), respectively.
Zhang et al. 2020	Randomized Control Trial	Recombinant Human Erythropoietin	Femoral Intertrochanteric fracture	141 patients; M = 64 (mean age 76.21 years, SD 7.90 years), F = 77 (mean age 79.16 years, SD 6.65 years)	Handgrip strength, ASM (appendicular skeletal muscle) index and post-operative stay and infection	In females, the handgrip strength during week one (13.9 ± 3.327 kg) became significantly higher in the intervention group than in the control group (9.30 ± 2.812 kg), and the difference was statistically significant (*p* < 0.05). During week two (13.212 ± 3.071) and week four (14.742 ± 3.375), the handgrip strength was consistently higher in the intervention group than in the control group (*p* ≤ 0.05). At the fourth week after EPO intervention, the ASM increment in the female and male intervention group (0.56 ± 0.43 kg,) was significantly higher than the ASM increment (0.24 ± 0.38 kg) in the control group over the fourth week.(*p* < 0.001). Infection rate in intervention group was significantly inferior and hospitalization state was significantly shorter.

KOVAL = walking ability scale; FAC = Functional Ambulatory Category; MMRI = modified Rivermead mobility index; BBS = Berg Balance Scale; MMSE = Mini-Mental State Examination; K-GDS = Korean version of the geriatric depression scale; EQ-5D = Euro quality-of-life questionnaire 5-dimensional classification; K-MBI = Korean modified Barthel index; K-IADL = the Korean instrumental activity of daily living.

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
