# Peer review of "Rehabilitation Strategies for Patients with Femoral Neck Fractures in Sarcopenia: A Narrative Review"

_jcm, 2020, doi:10.3390/jcm9103115_

Round 1
Reviewer 1 Report
This review summarized the effects of physical therapy with or without nutritional supplements on indicators of sarcopenia for femoral neck fracture patients. The issue addressed in this review seems to be crucial as the patients after hip fracture often cause or aggravate sarcopenic status and it should be treated with comprehensive approach including rehabilitation, nutrition and pharmacological measures.
However, there are several concerns should be addressed before publication.
#1 Overall, this manuscript needs to be edited by native English speaker thoroughly.
#2 P1 L38-42: The EWGSOP launched new diagnostic algorithm for sarcopenia in 2019(EWGSOP2, Cruz-Jentoft AJ, Bahat G, Bauer J, et al. Sarcopenia: revised European consensus on definition and diagnosis. Age Ageing. 2019;48(1):16-31). This statement should be revised based on the EWGSOP2 criteria.
#3: P2 L49-50: Malnutrition is partially overlapped, but different concept form sarcopenia. Please differentiate these two concept clearly.
#4: P2 L57: The term “of life” makes no sense.
#5: P2 L67-68: The aim of the study is a bit different from the literature search actually performed. The term used in the literature search were “rehabilitation” and “hip fracture” and “sarcopenia” and in fact, some studies included non-sarcopenic patients. I guess “The aim of this review is “to evaluate the impact of rehabilitation with or without other intervention including nutritional supplementation and pharmacological therapy on indicators of sarcopenia for FNF patients” may be better described the study’s purpose.
#6: P6 L102-113: Similar to #2, new EWGSOP2 criteria should be used in this section. In addition, new AWGS criteria (AWGS2019) in 2019 should also be cited and described in detail(Chen L-K, Woo J, Assantachai P, et al. Asian Working Group for Sarcopenia: 2019 Consensus Update on Sarcopenia Diagnosis and Treatment. J Am Med Dir Assoc. 2020;21(3):300-307.e2.).
#7 P6 L125-133: This paragraph is very confusing. The primary and secondary outcome measures are generally described in one paper and these indicators from several studies should not be integrated into one sentence. Please avoid using “primary outcome” and “secondary outcome” in this context. Instead, “outcome measures used in the eligible studies” is preferred.
#8 P8 L205-208: Similar to #7, Using “primary outcome” and “secondary outcome” should be avoided.
#9 P8 L234: There may be a typo: “in in”
#10 P9 L275: “However, Lim et al [22] considered this…” What does “this” stands for? please clarify.
#11 P9 L279-283: This sentences need appropriate citation.
#12 P9 L293: “… that improvements in diet among older adults at risk, ...” At risk of what? Please clarify.
#13 References should follow the author guideline. For example, the title of ref.14 should be written as small letter except for first letter. Please check thoroughly.
Author Response
This review summarized the effects of physical therapy with or without nutritional supplements on indicators of sarcopenia for femoral neck fracture patients. The issue addressed in this review seems to be crucial as the patients after hip fracture often cause or aggravate sarcopenic status and it should be treated with comprehensive approach including rehabilitation, nutrition and pharmacological measures.
However, there are several concerns should be addressed before publication.
Q1) Overall, this manuscript needs to be edited by native English speaker thoroughly.
A1) Thank you for your comment. As you suggested, English editing has been edited
Q2) P1 L38-42: The EWGSOP launched new diagnostic algorithm for sarcopenia in 2019(EWGSOP2, Cruz-Jentoft AJ, Bahat G, Bauer J, et al. Sarcopenia: revised European consensus on definition and diagnosis. Age Ageing. 2019;48(1):16-31). This statement should be revised based on the EWGSOP2 criteria.
A2) Thank you for your suggestion. Sarcopenia is defined as an acute or chronic disease based on low levels of measures for three patameters: muscle strength, muscle quanitity/quality and physical performance.
Q3) P2 L49-50: Malnutrition is partially overlapped, but different concept form sarcopenia. Please differentiate these two concepts clearly.
A3) Protein malnutrition is one the commonest factor associated with femoral neck fracture. Following your suggestion, we wrote: “Among these different comorbidities, apart from Sarcopenia, protein-energy malnutrition has been reported 20% to 85 %, depending on age and gender”.
Q4) P2 L57: The term “of life” makes no sense.
A4) Thank you, we corrected it.
Q5) P2 L67-68: The aim of the study is a bit different from the literature search actually performed. The term used in the literature search were “rehabilitation” and “hip fracture” and “sarcopenia” and in fact, some studies included non-sarcopenic patients. I guess “The aim of this review is “to evaluate the impact of rehabilitation with or without other intervention including nutritional supplementation and pharmacological therapy on indicators of sarcopenia for FNF patients” may be better described the study’s purpose.
A5) We agree and change the sentence.
Q6) P6 L102-113: Similar to #2, new EWGSOP2 criteria should be used in this section. In addition, new AWGS criteria (AWGS2019) in 2019 should also be cited and described in detail(Chen L-K, Woo J, Assantachai P, et al. Asian Working Group for Sarcopenia: 2019 Consensus Update on Sarcopenia Diagnosis and Treatment. J Am Med Dir Assoc. 2020;21(3):300-307.e2.).
A6) We corrected the cut off values and definitions of Sarcopenia with the more recent ones and updated the citations.
L 102-118: “Various definitions of Sarcopenia have been developed by different International Consensus Panels, (Asian Working Group on Sarcopenia, European Working Group on Sarcopenia in Older People, Foundation of the National Institute of Health, International Working Group on Sarcopenia) each defining cut-off values from mobility limitation measures (Appendicular Skeletal Mass Index, Grip Strength, HandgGrip Strength, Walking SpeedPhysical Performance). In our review, the authors specifically used the diagnostic criteria included in the ASWGS [21-25] and EWGSOP [1, 16, 19, 20].
The EWGSOP defines Sarcopenia when ASM is less than 20 kg for men and 15 kg for women, ASM/height2 is less than I is more than 78.90 kg/m2 for men and 56.370 kg/m2 for women (muscle quantity), HandgGrip strength is less than 2730 kg for men and 1620 kg for women, chair stand >15s for five rises (muscle strength), and Gait speed is no more than 0.8 m/s, Short Physical Performance Battery (SPPB) is less than 8 points, Timed Up and Go (TUG)Test is less than 20 s and 400 m walk test is completed in more than 6 minutes or not completed at all (physical performance) [1].
AWGS criteria include decreased handgrip strength (males < 286 kg, females < 18 kg), physical performance evaluated with gait speed (≤ 0.8 m/s or 5-time chair stand test: ≥12 s or Short Physical Performance Battery: ≤9), and loss of muscle mass, indexed by appendicular skeletal muscle mass (ASM) divided by height squared evaluated through Dual-energy X-ray absorptiometry (M: <7.0 kg/m2, F: <5.4 kg/m2) or Bioelectrical impedance analysis (M: <7.0 kg/m2, F: <5.7 kg/m2) (ASM/height2: males < 7.0 kg/m2, females < 5.7 kg/m2) [25].
Q7) P6 L125-133: This paragraph is very confusing. The primary and secondary outcome measures are generally described in one paper and these indicators from several studies should not be integrated into one sentence. Please avoid using “primary outcome” and “secondary outcome” in this context. Instead, “outcome measures used in the eligible studies” is preferred.
A7) Although we wanted to respect the authors’ intent to differentiate between primary and secondary outcomes in the studies, we understand that this may be confusing in the review. Anyway, not all the studies evaluated the same outcomes, so we had to list them.
Q8) P8 L205-208: Similar to #7, Using “primary outcome” and “secondary outcome” should be avoided.
A8) Thank you, we corrected.
Q9) P8 L234: There may be a typo: “in in”
A9) Thank you, we corrected the typo.
Q10) P9 L275: “However, Lim et al [22] considered this…” What does “this” stands for? please clarify.
A10) Thank you. We corrected as it follows: “However, Lim et al. [22] considered poor ambulatory independence as predictor factor for worst results in the evaluated outcome.”
Q11) P9 L279-283: This sentences need appropriate citation.
A11) We added the citation at the end of the paragraph.
Q12) P9 L293: “… that improvements in diet among older adults at risk, ...” At risk of what? Please clarify.
A12) Thank you, we clarified it: “In literature, less it is reported about the role of diet in older age, although there is an evidence that improvements in diet among older adults at risk of developing sarcopenia.”
Q13) References should follow the author guideline. For example, the title of ref.14 should be written as small letter except for first letter. Please check thoroughly.
A13) Thank you, we corrected them.
Reviewer 2 Report
A narrative review of 7 studies investigating sarcopenia following FNF which is comprehensive and sound. Requires some minor English editing.
Author Response
Q1) A narrative review of 7 studies investigating sarcopenia following FNF which is comprehensive and sound. Requires some minor English editing.
A1) Thank you for your comment. As you suggested, English editing has been edited.
Reviewer 3 Report
Dear Editor,
With interest, I reviewed the literature review by Avola et al., titled “Rehabilitation Strategies for patients with femoral 2 neck fractures in sarcopenia: a narrative review”. It is a very important topic, as many patients do not undergo proper postoperative rehabilitation and management. However the literature review suffers from a few flaws that I state below:
General comments:
- Not sure what your criteria are for the number of keywords, but there are nine some repeating the title.
Title needs rewording. Maybe: “Rehabilitation Strategies for sarcopenic patients with femoral neck fractures: a narrative review”.
MAJOR:
If the authors included those studies that had access to their data etc., why did not they do a meta-analysis; which is much more reliable than a simple literature review. Why did you not conduct a classic systematic review with registration etc.?
Please add one or two sentences of the conclusions of the authors to the end of sections you introduce the studies. Also, a sentence on the strengths and limitations of the studies is most welcome.
Abbreviations like ASMI have not been presented in full. The same applies to K-IDAL, K-FRAIL, KOVAL, FAC and ….
L108: ALM/h2 references to EWGSOP and EWGSOP2 are wrong. They are 7023 and 5.5 of top of my head. You use the non-standard terms like ASMI, ASM index and ASM throughout. Please correct.
L264 “Physical inactivity is clearly 264 linked to losses of muscle mass and strength” should be “Physical inactivity was negatively linked to losses of muscle mass and strength”.
The authors use the names of the first author of “study” etc. interchangeably. If they are keen to discuss each study individually, I highly recommend naming them as Study 1 to Study 7, and introducing them so in Table one. The current format is very confusing for an audience that is supposed to read the paper once to grasp the core of it. This makes cross-referencing much easier.
Please replace the term “results measures” with “outcomes” throughout.
L209: Please state the number of sarcopenic vs non sarcopenic for the this study and the previous ones. The same applies to controls and cases in each study.
L212-214: You are comparing sarcopenic vs sarcopenic. One of them must be non-sarcopenic.
L228-9 ALM corrected for height2, BMI or without adjustment?
You do not present any data on mortality based on my quick check, and yet discuss in in this section. This is a big flaw.
I did not check the references and the seven papers included in this literature review, but I trust the Editors will do so.
MINOR
Table 1 should be presented in landscape format, as it is hard to read.
A table indicating which study used which sarcopenia definition would be helpful. This can be added to Table 1.
I reckon the authors have not really done a great job searching what really is out there in terms of available literature. Here is a list of relevant papers (some of them valuable references), by Google Scholar:
https://scholar.google.com.au/scholar?hl=en&as_sdt=0%2C5&as_vis=1&inst=4522501446918153378&q=neck+of+femur+fracture+management+rehabilitation+nutrition&btnG=
Author Response
Dear Editor,
With interest, I reviewed the literature review by Avola et al., titled “Rehabilitation Strategies for patients with femoral 2 neck fractures in sarcopenia: a narrative review”. It is a very important topic, as many patients do not undergo proper postoperative rehabilitation and management. However the literature review suffers from a few flaws that I state below:
General comments:
Q1) Not sure what your criteria are for the number of keywords, but there are nine some repeating the title.
A1) Thank for your comment. As reported in Instruction for Authors, from “three to ten pertinent keywords need to be added after the abstract”. According to your suggestions, keywords were changed.
Q2) Title needs rewording. Maybe: “Rehabilitation Strategies for sarcopenic patients with femoral neck fractures: a narrative review”.
A2) Thank for your kind suggestion. The title has been modified.
MAJOR:
Q3) If the authors included those studies that had access to their data etc., why did not they do a meta-analysis; which is much more reliable than a simple literature review. Why did you not conduct a classic systematic review with registration etc.?
A3) Thank you for your observation. We could not administer a meta-analysis because the outcomes evaluated thorough the studies were not always heterogeneous. Furthermore, the high presence of observational and non-randomized trial would have created several biases.
Q4) Please add one or two sentences of the conclusions of the authors to the end of sections you introduce the studies. Also, a sentence on the strengths and limitations of the studies is most welcome.
A4) Thank you for your kind suggestion. When they were not present, we added limitations and conclusions of the studies.
L144-149: “Limitations of this study were the short time after which the outcome were evaluated (after 2-weeks of interventions), the exclusion of several patients before the start of the treatment and the use of sarcopenia classification itself may have affected the group allocation. Even though the results of Study 1 suggest that FIRM was effective for short-term functional recovery in older patients with or without sarcopenia who have suffered fragility hip fracture, further research comparing it with conventional therapy is needed.”
L165-167: “Anyway, several limitations were caused by the assessment of Sarcopenia in patients soon after the surgery, the time of follow-up that in fragile patients may be longer and the lack of a control group following conventional rehabilitation.” Conclusions were already written.
L182-185: “Limitations of Study 3 were the high amount of drop outs after hospital discharge, it was carried out in only one center and the number of the sample was not sufficient to significantly represent sub-groups with different cognitive function, hip fracture and hip operation type. “
L206: “Limitation of this study was the small sample size”.
L 234-237: “ Even though it was performed on a small sample size, data emerging from this study showed a good impact of this combined intervention on function and disability in hip fracture patients after 2 months of treatment. Essential amino acid supplementation induced considerable improvements in the sarcopenic sub-population of the study.”
L259-267: “This study has a number of limitations. Patients received rehabilitation five days a week. It would be interesting to see whether participation in a programme of resistance exercises during patients’ stay at a rehabilitation centre improves the functional results reported. Authors could not do any follow-up of patients after discharge to assess whether the benefits obtained were maintained. Diagnostic criteria for sarcopenia proposed by the EWGSOP are difficult to apply in patients with hip fractures admitted to rehabilitation units, because most of the patients are unable to walk when they arrive.Although these limitations, this research has some important strengths. Due to the characteristics of the patients included, this study could be representative of the geriatric population admitted to rehabilitation centres.”
Q5) Abbreviations like ASMI have not been presented in full. The same applies to K-IDAL, K-FRAIL, KOVAL, FAC and ….
A5) Thank you. The sentences including ASMI were changed.
Q6) L108: ALM/h2 references to EWGSOP and EWGSOP2 are wrong. They are 7023 and 5.5 of top of my head. You use the non-standard terms like ASMI, ASM index and ASM throughout. Please correct.
A6) Thank you. This correction has been made following the correction of reviewer 1. (Q2 and Q6)
Q7) L264 “Physical inactivity is clearly 264 linked to losses of muscle mass and strength” should be “Physical inactivity was negatively linked to losses of muscle mass and strength”.
A7) Thank you for the correction. It sounds much better.
Q8) The authors use the names of the first author of “study” etc. interchangeably. If they are keen to discuss each study individually, I highly recommend naming them as Study 1 to Study 7, and introducing them so in Table one. The current format is very confusing for an audience that is supposed to read the paper once to grasp the core of it. This makes cross-referencing much easier.
A8) We agree. Thank you for your suggestion. We changed the text.
Q9) Please replace the term “results measures” with “outcomes” throughout.
A9) Thank you, we have already changed it following the correction of reviewer 1 (Q7 and Q8).
Q10) L209: Please state the number of sarcopenic vs non sarcopenic for the this study and the previous ones. The same applies to controls and cases in each study.
A10) Thank you for the suggestion. We added the number of sarcopenic- non sarcopenic and controls and cases in each study.
Q11) L212-214: You are comparing sarcopenic vs sarcopenic. One of them must be non-sarcopenic.
A11) We compared sarcopenic patients in two different groups (A and B).
Q12) L228-9 ALM corrected for height2, BMI or without adjustment?
A12) ALM was corrected for height2. We added it in the text.
Q13) You do not present any data on mortality based on my quick check, and yet discuss in in this section. This is a big flaw.
A13) Mortality wasn’t a topic of our review, so we prefer to delete the term it in this section. Thank you for your attention.
Q14) I did not check the references and the seven papers included in this literature review, but I trust the Editors will do so.
A14) We agree.
MINOR
Q15) Table 1 should be presented in landscape format, as it is hard to read.
A15) We followed the guidelines of your journal. If you prefer, we can present in landscape format.
Q16) A table indicating which study used which sarcopenia definition would be helpful. This can be added to Table 1.
A16) We agree. We added to Table 1.
Q17) I reckon the authors have not really done a great job searching what really is out there in terms of available literature. Here is a list of relevant papers (some of them valuable references), by Google Scholar:
https://scholar.google.com.au/scholar?hl=en&as_sdt=0%2C5&as_vis=1&inst=4522501446918153378&q=neck+of+femur+fracture+management+rehabilitation+nutrition&btnG=
A17) Thank for your comment. As described in methods, a review of PubMed, Cochrane Library and PEDro was performed, and after the screening 7 manuscript were included in the study. We are sure that it is not possible to report all the literature data but, in our opinion, the main and more recent finding were investigated in our narrative review.
Round 2
Reviewer 1 Report
The authors' effort substantially improve the manuscript. I just point out only one typo:
L107: "In our review, the authors specifically used the diagnostic criteria included in the ASWGS [21-25] and EWGSOP [1, 16, 19, 20]" should be used "AWGS".
Author Response
L107: "In our review, the authors specifically used the diagnostic criteria included in the ASWGS [21-25] and EWGSOP [1, 16, 19, 20]" should be used "AWGS".
Dear reviewer, the typo has been correct. Thank you for your comments
Reviewer 3 Report
Dear Editor, I only reviewed the responses of the authrs to my points, and not the text, as the changes oddly have not been made in colour or using Track changes. I trust you and team will check them in the text.
I am happy with the authors responses; however, as mentioned there are other papers if the search is extended.
I also did not check for plagiarism, self-citations etc.
You may seek feedback from a third reviewer.
Author Response
Thank you for your revision